# Detection of Tobacco Bacterial Wilt Caused by *Ralstonia solanacearum* by Combining Polymerase Chain Reaction with an α-Hemolysin Nanopore

**DOI:** 10.3390/nano13020332

**Published:** 2023-01-13

**Authors:** Ying Wang, Yusen Li, Xin Zhou, Wenna Zhang, Shusheng Zhang, Dongmei Xi

**Affiliations:** 1Shandong Provincial Key Laboratory of Detection Technology for Tumor Markers, College of Life Science, Linyi University, Linyi 276005, China; 2Shandong (Linyi) Institute of Modern Agriculture, Zhejiang University, Linyi 276000, China

**Keywords:** α-HL nanopore, *Ralstonia solanacearum*, polymerase chain reaction, lambda exonuclease, tobacco bacterial wilt

## Abstract

Tobacco bacterial wilt is a serious disease caused by the soil-borne bacterium *Ralstonia solanacearum* (*R. solanacearum*). Herein, a rapid and purification-free α-hemolysin (α-HL) nanopore-sensing strategy based on polymerase chain reaction (PCR) and lambda exonuclease digestion was established to detect *R. solanacearum*. A 198-nucleotide-long single-stranded DNA was obtained via asymmetric PCR or the lambda exonuclease-mediated digestion of the PCR product. The DNA fragment produced unique long-lived, current-blocking signals when it passed through the α-HL nanopore. This sensing approach can allow for the determination of *R. solanacearum* in tobacco samples and can be conveniently extended to other DNA monitoring because of the extremely wide range of PCR applications.

## 1. Introduction

The rapid clinical or agricultural detection of pathogenic DNA usually requires polymerase chain reaction (PCR) amplification [1] to obtain double-stranded DNA (dsDNA) products, which are then identified using fluorescence-based [2] or lateral flow strip methods [3]. Generally, the fluorescence-based methods require applying dyes or modified luminescent and quenched groups to the probe and corresponding instruments [4,5]. Reaction conditions, such as temperature, solvent, acidity, and fluorescence quenchers, affect the intensity of fluorescent signals [6]. The lateral flow strip method is simple and rapid; however, it suffers from shortcomings, such as low throughput and high cost, and its accuracy is directly related to the specificity of the capture agent [7,8]. Therefore, the development of an alternative rapid and label-free detection method with good accuracy is necessary [9].

Nanopore has emerged as an attractive single-molecule tool that offers rapid, low-cost, and label-free analysis [10]. The ionic current fluctuation through the nanopore caused by an analyte binding can be utilized to reveal the identity and concentration of the analyte [11]. To date, the nanopore technique has been widely developed for the detection of various analytes including DNA [12], microRNA [13], peptides [14,15], small molecules [16], metal ions [17], and biomacromolecules [18,19]. Nanopore sequencing evolved from α-HL biological nanopores to MinION solid-state nanopores developed by the Oxford Nanopore Corporation and have been used to analyze human, cancer, pathogen, and other samples [20,21,22]. Considering that PCR is widely used to amplify DNA, the combination of PCR and a nanopore sensor could greatly expand the application range of nanopore technology. Recently, SiNx solid-state nanopores with a diameter >5 nm have been used as downstream sensors for PCR products, and clinical samples of group A *Streptococcus* have been successfully identified [23]. However, it remains challenging to distinguish the blocking current signals of PCR products from those generated by the molecules in PCR buffer.

Tobacco bacterial wilt is a serious soil-borne disease caused by *Ralstonia solanacearum* (*R. solanacearum*) [24]. It blocks vascular bundles and causes wilting in plants until the entire plant dies [25]. In this study, we present a facile strategy for monitoring the DNA of *R. solanacearum* by coupling PCR with a nanopore sensor. In general, the size of the channel diameter is the dominant factor that affects the distinguishing ability and application scope of the nanopore. Herein, α-hemolysin (α-HL), a well-studied bacterial-derived toxin with the narrowest constriction of 1.4 nm [26,27,28,29], was utilized for analysis. A single-stranded DNA (ssDNA) with 198 nucleotides was obtained via asymmetric PCR or the lambda exonuclease-mediated degradation of the PCR product. In the presence of a salt concentration gradient, the ssDNA generated remarkably long current-blocking signals with a high capture rate when passing through the α-HL nanopore. Notably, the unique long-lived signals were easily distinguished from those generated by materials in the PCR reaction buffer. This method was used to detect *R. solanacearum* in tobacco field samples without labeling or sample purification, and the results obtained were consistent with those obtained via sequencing. Due to the wide application of the PCR method, this strategy combining PCR and a nanopore sensor has great potential in nucleic acid monitoring applications.

## 2. Experimental Section

### 2.1. Chemicals

1,2-Diphytanoyl-sn-glycero-3-phosphocholine (chloroform, ≥99%) was purchased from Avanti Polar Lipids Inc. (Alabaster, AL, USA). α-HL (lyophilized powder from Staphylococcus aureus) and decane (anhydrous, ≥99%) were purchased from Sigma-Aldrich (St. Louis, MO, USA). Trans DNA Marker I was purchased from Beijing Transgen Biotechnology Co., Ltd. (Beijing, China). Lambda exonuclease was purchased from ThermoFisher Scientific Ltd. (Shanghai, China). Rapid Taq Master Mix was purchased from Vazyme Biotech Co., Ltd. (Nanjing, China). A Rapid DNA Extraction and Detection Kit (KG203) and Universal DNA Purification kit (DP214) were purchased from Tiangen Biochemical Technology Co., Ltd. (Beijing, China). All other reagents were of analytical grade and used without further purification. One positive and five field-collected tobacco samples were kindly provided by the Laboratory of Plant Protection Research Center at the Tobacco Research Institute of the Chinese Academy of Agricultural Sciences. Cyto-1F (5’-TTCCAGTATCTCAGCCCGGA-3’), Cyto-1R (5’-GGTCACCTTGAAGACGCCAG-3’), and p-Cyto-1R (5’-pGGTCACCTTGAAGACGCCAG-3’) were synthesized by Sangon Biotech Co., Ltd. (Shanghai, China). The 198-nucleotides sequence (strand A) was synthesized and purified by Takara Bio Inc. (Dalian, China).

### 2.2. Apparatus

DNA concentration was determined using NanoDrop One (Thermo Scientific, Waltham, MA, USA). PCR was performed using a mini-amplifier (Thermo Scientific, Waltham, MA, USA). The electrophoresis apparatus DYY-8C (Liuyi, Beijing, China) and a horizontal electrophoresis cell, DYCP-31DN (Liuyi, Beijing, China), were used for gel electrophoresis. Gel images were captured using a Universal Hood II Imaging System (Bio-Rad, Hercules, CA, USA). Current was measured using a patch-clamp amplifier (Axon 200 B, AXON, Scottsdale, AZ, USA) and converted into data using a DigiData 1440A converter (Molecular Devices, San Jose, DE, USA).

### 2.3. DNA Extraction and PCR

The genomic DNA of *R. solanacearum* (1.5 mL bacterial culture) or the total DNA of tobacco samples infected with *R. solanacearum* (about 0.5 g) were extracted and purified using the rapid DNA extraction and detection kit. The genome was used as the template for PCR. The target sequences of *R. solanacearum* were amplified in a 20 μL reaction volume containing 10 μL 2× Rapid Taq Master Mix, 1 μL Cyto-1F (10 μM), 1 μL p-Cyto-1R (10 μM) (or 0.25 μL Cyto-1R (10 μM) in asymmetric PCR), 1 μL genomic DNA, and 7 μL H_2_O [30]. Thermal cycling consisted of denaturation at 94 °C for 1 min, 30 cycles at 94 °C for 30 s, 60 °C for 30 s, and 72 °C for 30 s, and a final extension at 72 °C for 5 min. The PCR products were separated using a 2% agarose gel stained with GelRed dye and visualized using the Universal Hood II Imaging System. The positive PCR products of the tobacco samples were submitted to Sangon Biotech Co., Ltd. (Shanghai, China) for sequencing.

### 2.4. Digestion by Lambda Exonuclease

In the presence of lambda exonuclease, the PCR product was degraded stepwise from its 5-phosphorylated end into ssDNA. The reaction was carried out using 50 µL 1× reaction buffer (67 mM glycine-KOH, 2.5 mM MgCl_2_ 0.01% (*v/v*), Triton X-100 (pH 9.4 at 25 °C)). The PCR product (10 μL) and lambda exonuclease (1 U) were then added to the reaction solution, and the reaction was performed at 25 °C for 5 min, followed by incubation at 85 °C for 5 min to inactivate lambda exonuclease.

### 2.5. Nanopore Electrical Recording and Data Analysis

A lipid bilayer membrane was formed by spanning a 150 μm orifice in a Delrin bilayer cup that was partitioned into two chambers: *cis* and *trans*. Both chambers were filled with 1 mL of buffer (1 M KCl, 1 mM EDTA, and 25 mM HEPES; pH 7.0). α-HL was injected close to the hole in the *cis* chamber, and insertion was confirmed by a definite jump in the current values. When a stable single-pore insertion was detected, DNA was added to the *cis* chamber immediately adjacent to the aperture. A positive potential of 150 mV was applied to the *trans* chamber of the detection cell using an Ag/AgCl electrode. The conditions for nanopore electrical recording using gradient salt concentrations were similar to those of equal salt concentrations; however, the concentration and voltage were changed. Both chambers were filled with 1 mL of buffer (*cis*: 0.75 M KCl, 1 mM EDTA, 25 mM HEPES, pH 7.0; *trans*: 3 M KCl, 1 mM EDTA, 25 mM HEPES, pH 7.0). A positive potential of 100 mV was applied to the *trans* chamber of the detection cell using an Ag/AgCl electrode. Nanopores were measured at 25 ± 2 °C. After a nanopore electrical recording test, the ssDNA in the two chambers were extracted and purified using the Universal DNA Purification kit and amplified with the PCR method according to Section 2.3 of the paper.

Current was measured using the patch-clamp amplifier and converted into data using a DigiData 1440A converter. The signals were filtered at 5 kHz and acquired at a sampling rate of 100 kHz using PClamp 10.6 (Axon Instruments, Foster City, CA, USA). Finally, data were analyzed using MATLAB (R2013b, MathWorks, Natick, MA, USA), OriginLab 9.0 (OriginLab Corporation, Northampton, MA, USA), and software programmed by Long’s group [31]. The original data of the Axon Binary Data files (ABF) of the nanopore experiment were converted into TXT files using Cutting software programmed by Long’s group [31]. The TXT files were processed using MATLAB, and the current signals through the nanopore were screened. The screened data were imported into OriginLab 9.0 for function fitting, and scatter plots and blocking degrees were obtained. Values in the error bars were calculated using the “Statistics” function of OriginLab 9.0.

## 3. Results

### 3.1. Feasibility of Detecting R. solanacearum by Combining Asymmetric PCR with an α-HL Nanopore

This strategy was used to detect *R. solanacearum* by combining asymmetric PCR with nanopore sensing. As shown in Figure 1, genomic DNA was extracted from the *R. solanacearum* cells. The mixture of single- and double-stranded DNA was obtained via asymmetric PCR using the genomic DNA of *R. solanacearum* as the template and the primer pair Cyto-1F/Cyto-1R. The nanopore formed by α-HL was used as the sensor element. The diameter of ssDNA is smaller than the inner diameter of α-HL, and the diameter of dsDNA is larger than the inner diameter of α-HL. Therefore, ssDNA can pass through the α-HL nanopore in the electric field, causing characteristic current blocks. The passage of dsDNA through the α-HL nanopore is difficult without a specific guide strand at its end [32,33].

The PCR product contains dsDNA, and the two single strands are called strands A and B. Here, strand A (red in Figure 1) was produced via PCR amplification using Cyto-1F as a primer and its complementary strand, B, as a template. Similarly, strand B (green in Figure 1) was generated using Cyto-1R as a primer and complementary strand A as a template. To verify whether asymmetric PCR products were generated, we first monitored amplicons using 2% agarose gel electrophoresis. As shown in Figure 1A, the primer pair Cyto-1F/Cyto-1R was used to amplify the genomic DNA of *R. solanacearum* at a volume ratio of 4:1, and a mixture of strand A (198 nucleotides) and dsDNA (198 base pairs) was obtained (Lane 3). In addition, a negative control was obtained via PCR amplification using ultrapure water as a template (Lane 4).

Next, in the presence of a positive potential of 150 mV, a nanopore-based assay was employed to analyze asymmetric PCR products in 1 M KCl (*cis*/*trans*). In the negative control experiment, several block events appeared in the current trace (Figure 1B, top). Scatter plots showed that these blocking events had blocking degrees within a range of 0.2–0.96 (Figure 1C), similar to the results reported by King’s group [23]. The PCR mixture contained replicase, bovine serum albumin, and other molecules, and these molecules produced translocation signals in the nanopore assay. Some of these signals also may be caused by collisions.

In the presence of asymmetric PCR products with both single-stranded and double-stranded DNA, we observed several markedly prolonged events (Figure 1B, bottom) that were easily discriminated from those generated by the negative control. For statistical analysis, I_0_ and I were designated as the open pore current and blockage current, respectively. The scatter plot showing current blockage and duration yielded two populations upon the addition of asymmetric PCR products (Figure 1D), of which P2 had characteristics similar to those of the negative control group (Figure 1C). Notably, the newly emerged population, P1, had a wide distribution of dwell time, from 22.49 to 2067.07 ms. These signals are clearly different from those generated by the molecules in the PCR mixture, initially confirming the translocation of the 198-nucleotide-long DNA strand through the nanopore.

### 3.2. Enhancing the Performance of the PCR–α-HL Strategy Using Lambda Exonuclease and Gradient Salt Concentration

Although the presence of *R. solanacearum* could be detected by combining asymmetric PCR with a nanopore, the capture rate of the ssDNA was rather low, and the α-HL nanopore was frequently blocked, sometimes reversibly and, more often, irreversibly, probably because the dsDNA with blunt ends could not pass through the nanopore and blocked it.

To address those issues, we aimed to obtain only one strand of the PCR product via enzymatic digestion (Figure 2). The genomic DNA of *R. solanacearum* was amplified via PCR using the primers Cyto-1F/p-Cyto-1R. The 5’-terminal hydroxyl group of the primer p-Cyto-1R was linked to a phosphate group, so the PCR product contained a phosphate group at the 5’-terminus of strand B, and no phosphate group was present at the 5’-end of strand A. Lambda exonuclease is a 5’-3’ exonuclease that catalyzes the stepwise removal of nucleotides at the 5’-phosphorylated strands of dsDNA, with low activity for non-phosphorylated substrates. Therefore, lambda exonuclease degraded strand B, whereas strand A remained unaffected. It passed through the α-HL nanopore and produced characteristic current blocks.

As shown in Figure 2A, the genomic DNA of *R. solanacearum* was amplified using the primer pair Cyto-1F/p-Cyto-1R, and a bright band was observed on a 2% agarose gel (lane 2). Lambda exonuclease degraded strand B of the dsDNA, and ssDNA (strand A) was produced. The brightness of the band corresponding to strand A was significantly lower than that of the PCR products (Lane 3).

Asymmetric salt concentration across the α-HL nanopore can significantly improve the capture rate of target DNA [34]. We speculated that this strategy could also be effective in this assay. As expected, the unpurified lambda exonuclease-digested PCR products containing strand A in asymmetric KCl solutions (0.75 M/3 M, *cis*/*trans*) across the nanopore produced significantly more current blocks than those in 1 M KCl (*cis*/*trans*) conditions (Figure 2B). In addition, the lambda exonuclease-treated PCR product (strand A) was purified by the Universal DNA Purification kit. Strand A was synthesized and purified by Takara Bio Inc. The duration of both kinds of strand A was decreased as the voltage increased. At the end of the translocation test, the synthesized strand A in the two chambers were purified by the Universal DNA Purification kit, which was used as the template for the PCR. The results show that strand A was amplified by the *trans* chamber and the *cis* chamber. These experimental results verified that strand A was translocated to the *trans* chamber (Appendix A).

Importantly, because of the improved capture rate, this strategy recognized the ssDNA of *R. solanacearum* more efficiently. In contrast to the scatter plot of the negative control experiment (Figure 3A,C), a new cluster of multiple signals appeared in the scatter diagram of the target DNA (Figure 3D). In this population, the current blockage (I/I_0_) was greater than that of the negative control. Statistical analysis showed that the I/I_0_ of the higher population was fitted to the Gauss distribution with a value of 0.79 ± 0.0057 (Figure 3E). The duration lay within a range of 20.11–7232.85 ms and statistically fitted the exponential function with a value of 165.50 ± 16.15 ms (Figure 3F). These data, integrated with the results of electrophoresis, suggested that the population was generated by the translocation of strand A through the nanopore and can serve as output DNA signatures for *R. solanacearum* analysis. The results indicate that the performance of the PCR-α-HL strategy was effectively improved using lambda exonuclease and gradient salt concentration.

### 3.3. Application of This Method to Tobacco Fields

PCR and DNA sequencing are the most common methods used to identify pathogens. To verify the applicability of this strategy in field samples, one positive and five field samples were tested using this approach and DNA sequencing (Appendix A). The results of both methods indicated that the positive sample and four field samples were infected with *R. solanacearum*. The detection efficiency of the two methods is consistent, indicating that the PCR–α-HL nanopore strategy can be used to detect *R. solanacearum* in tobacco fields (Table 1).

## 4. Conclusions

In this study, we established a rapid, purification-free, and label-free strategy for detecting *R. solanacearum* by combining PCR with an α-HL nanopore. Current blocking produced by the translocation of ssDNA was clearly distinguishable from that generated by molecules in the PCR mixture. To the best of our knowledge, this is the first report of the PCR–α-HL-nanopore-based detection of *R. solanacearum* and its tobacco field samples, which lays a solid foundation for establishing an early warning mechanism for tobacco bacterial wilt. In the future, the coupling of isothermal amplification methods with nanopore sensors will be further studied, and nanopore-based methods will be used for the multiple detection of pathogens in certain samples.

## Data Availability

The data presented in this study are available in the main manuscript and in the Appendix A.

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
