# Peer review of "Detection of Tobacco Bacterial Wilt Caused by Ralstonia solanacearum by Combining Polymerase Chain Reaction with an α-Hemolysin Nanopore"

_nanomaterials, 2023, doi:10.3390/nano13020332_

Round 1

Reviewer 1 Report

Wang et al., has presented nice piece of work. Author demonstrated rapid and purification-free α-hemolysin (α-HL) nanopore sensing strategy for detection of R. solanacearum using PCR and Lamda exonuclease.

Author needs some edit in the figure captions 1. Introduction can be elorated more on nanopore sequencing.

1. Description can be added on how exonuclases has bee attached to the nanopore?

Reviewer 2 Report

The authors described excellent methodology, but the data analysis part has to be elaborated to allow other research groups to repeat.

Reviewer 3 Report

This manuscript by Wang et al. reports on use of biological nanopore sensing for detection of a pathogenic bacterium-derived DNA.  The authors used α-hemolysin as a nanopore sensor to analyze PCR products of Ralstonia solanacearum.  The ionic traces showed many short pulses in the negative controls in 1 M KCl that were explained as due to the presence of small molecules like proteins.  When the bacteria were present, on the other hand, they observed ionic current signals with much wider pulse widths.  They obtained similar results when applying a salinity difference across the membrane, where they also observed enhanced capture rates.  Together with the agarose gel electrophoresis analyses, the overall results suggested the detections of the R. solanacearum-derived ssDNA by the ionic current measurements.  More importantly, the authors exploited the nanopore approach to real samples to show its feasibility for bacterial screening.

The work is well-organized and can be a good contribution to the field of biotechnology.  I will recommend publication of this paper after the authors address the following points by a revision:

1.       There is already Oxford nanopore technology that can perform amplicon sequencing of bacterial genomes.  The authors should explain the advantages of their approach over this well-established technology.

2.       Amplicon analysis was also reported using solid-state nanopores in King et al., ACS Sens. 7, 207 (2022).  This article can be cited in the introduction.

3.       Given the long translocation time of ssDNA, I wonder whether the biopolymers were able to pass through the nanopore or just moved back and forth.  In the latter case, the meaning of the capture rates becomes quite different.  To verify this point, it is recommended to check the presence of the DNA in the trans chamber after the measurements.

4.       Although the experiments were carefully designed to ensure the detections of ssDNA in the PCR products, it is better to test a case for a pure sample of synthetic DNA having the same sequence as the one for R. solanacearum.

5.       The capture rates of ssDNA vary among the real samples measured as shown in Figure S2.  Also, the number of DNA signals they got for each sample (Figure S4) does not seem to match with the capture rates.  Were there some additional issues like more often nanopore clogging when measuring the real samples?  How much time was spent on each measurement and how many pulses were obtained?

6.       Nanopore measurement results of the sample 5 should be shown in Figures S2-S4.

7.       The bin size and the range of I/I_0 should be the same for the four plots in Figure S4 to show it more clearly what is different among the four samples.

Round 2

Reviewer 3 Report

The manuscript has been improved greatly after the revision.  I now recommend its publication.